# Effects of Bentonite Addition on the Speciation and Mobility of Cu and Ni in Soils from Old Mine Tailings

Yongping Gao * and Xiaojun Li 

Shapotou Desert Research and Experiment Station, Northwest Institute of Eco-Environment and Resources, Chinese Academy of Sciences, Lanzhou 730010, China
* Correspondence: gaoyongping@nieer.ac.cn

**Abstract:** Bentonite has important applications in curbing soil heavy metal pollution. Jinchang city is an important industrial city in western China, and the mining activities in this area inevitably lead to the heavy metal pollution of soil; in particular, the maximum concentrations of Ni and Cu in the soil exceeded the standard by 30 and 25 times, respectively. In this study, we conducted soil cultivation experiments to study the effects of bentonite addition (BA) on the fractions of the heavy metals Ni and Cu in an old tailings area of Jinchang city. Surface soil (0–20 cm) samples were collected, and Ni and Cu fractionation was performed using the Tessier sequential extraction method. The results showed that residual (R) was the main fraction of Ni and Cu, which accounted for 53% and 57% of their total amounts, respectively. The fraction bound to organic matter (BM), bound to Fe-Mn oxides (BO), bound to carbonates (BC), and the exchangeable (E) accounted for 20% and 16%, 18% and 12%, 6%, and 7%, 3% and 4% of the total amounts, respectively. Their contents ranked in the order: R > BM > BO > BC > E. Each fraction of Ni and Cu decreased with increasing levels of BA. The proportion of E of Ni and Cu was considerably reduced, while the proportion of BO and R increased significantly following the BA. BA can significantly reduce the mobility of Ni and Cu. Our findings indicated that BA can significantly reduce the biological toxicity and mobilization of heavy metals in polluted soil, which can be recommended as a safe stabilizer for heavy metal pollution in soil.

**Keywords:** bentonite; heavy metals; heavy metal forms; organic matter; pH value



## 1. Introduction

With the progress of science and technology and the rapid development of modern industry, people's living standards have been significantly improved. However, the consequent heavy metal pollution has also become a serious environmental problem worldwide, and the heavy metal pollution in soil, in particular, has attracted more and more attention [1,2]. According to China's soil pollution investigation report, soil contamination by heavy metals is serious in some abandoned industrial and mining areas, with 17% of the heavy metals exceeding their permissible limits, and their contents were far beyond the standards [3]. Especially in the surrounding area of the mining district, pollution is becoming more and more serious. For example, in open-pit coal mining areas, the accumulation of Cr, Cu, Pb, and Zn in the soil is higher, but in copper mines, the accumulation of Cu, Zn, Pb, Cd, and Ni in the soil is higher [4].

Jinchang city is an important industrial city in western China, and the mining activities in this area inevitably lead to the heavy metal pollution of the soil. Liao et al. [5] reported that soil pollution by Ni and Cu, the main mineral products, is an incredibly serious issue. The maximum concentrations of Ni and Cu in the soil can reach 6102 mg/kg and 10321 mg/kg, which are 30 and 25 times higher than the values in the III level of "The Soil Environmental Quality Standard" (GB15618-1995), respectively, and 70% and 57% of soil's Ni and Cu concentrations exceed the permissible limits, respectively. Once entering into the soil, heavy metals diffuse and migrate slowly and cannot be degraded

by microorganisms. They are easy to transform into different chemical forms through the processes of dissolution, sedimentation, coagulation, complexation, and adsorption, etc. [6]. Heavy metals may affect human and animal health through the food chain, dust, and the pollution of surface and underground water when they accumulate in the soil to a certain extent, potentially causing great harm [7].

Currently, the total amount of heavy metals in soil is extensively used to evaluate the potential risks of polluted areas [8]. However, once heavy metals enter the soil, various physical and chemical effects will alter the morphology of heavy metals, mainly because soil heavy metals have different fractions with different availability [9]. Therefore, to evaluate the mobility, bioavailability, and toxicity of soil heavy metals, the fractions of heavy metals should be analyzed, rather than only analyzing their total contents [10,11]. There is no unified definition for the fraction classification of heavy metals; nevertheless, the most widely used method was proposed by Tessier et al. [12], who classified heavy metals into five fractions: exchangeable, bound to Fe-Mn oxides, bound to carbonates, bound to organic matter, and residual. The migration ability of heavy metals varies with fractions, in the order of: exchangeable > bound to carbonate > bound to Fe-Mn oxides > bound to organic matter; the residual fraction hardly migrates and is relatively stable [13]. The bioavailability of heavy metals is positively related to their mobility, and those with strong mobility and high bioavailability, such as the fractions bound to carbonates and exchangeable, are defined as effective forms. The transport ability of the bound to organic matter fraction and bound to Fe-Mn oxides fraction is weak, and they are defined as the potential effective fraction and can be changed into the effective fraction. The residual fraction is also defined as an unusable fraction due to its strong stability and low bioavailability. It is stable in nature and hard to release under normal conditions, remains stable in the soil for a long time, and is barely absorbed by plants. Therefore, the residual fraction has the least impact on the food chain in the whole soil ecosystem [14].

Curbing heavy metal pollution in soil has always been a key issue in restoration ecology [15–17]. At present, the methods for the remediation of heavy metal pollution in soil mainly include physical methods (deep ploughing method, soil replacement, etc.), chemical methods (in situ passivation technology, etc.), biological methods (phytoremediation, and plant barrier technology, etc.), and agronomic methods (adjustment of planting structure, etc.) [18,19]. Each method has its advantages and disadvantages and a range of practical applications [20]. Engineering measures are a practical and effective method to control soil heavy metal pollution, but this method requires massive amounts of labor and financial resources; there are also adverse factors such as land occupation, leakage, and environmental pollution in the process of soil replacement. Chemical remediation is easy to carry out; the heavy metal elements, however, still exist in the soil, which is easily activated and may cause secondary pollution. Agronomy biotechnological measures can effectively reduce the chance of heavy metals entering the human body through the food chain to a certain extent, and obtain better economic, social, and environmental benefits. However, these technologies are mainly in the earlier stage (laboratory experiments, field trials, and demonstration), lacking remediation practices for contaminated soils. Currently, the commonly used soil amendments mainly include silicon calcium material, phosphorus-containing materials, clay mineral, biochar, etc. [21], among which bentonite has been successfully used in soil improvement and wastewater treatment due to its advantages of a large specific surface area, strong ion exchange adsorption capacity, and strong stability [22]. Bentonite is a clay mineral with montmorillonite as its main component, which is also widely distributed worldwide with abundant reserves. Meanwhile, the addition of bentonite can improve soil nutrient availability and soil physicochemical properties, and it has high cation exchange capacity, strong water retention capacity, and good cohesion [23]. Lin et al. [24] found that bentonite addition can significantly increase soil pH, reduce the content of available nitrogen in the soil, available phosphorus, and organic matter, and ultimately impact the effective and exchangeable fractions of heavy metals in the soil. Moreover, the addition of bentonite can considerably enhance the solubility of Cd in the soil and its adsorption by

plants [10,25]. These findings are related to the change in heavy metal fractions in the soil. Therefore, it is of great significance to study the change in heavy metal fractions in the soil induced by the addition of bentonite for us to better understand the environmental effects of heavy metals and the remediation of heavy metal-contaminated soil.

In this study, the form changes of Ni and Cu following bentonite addition were investigated by collecting the soil samples polluted by heavy metals in the old Ni and Cu tailings area in Jinchang city. The main objectives are (1) to test the effects of bentonite addition on heavy metal speciation in soil; (2) to evaluate the influence of bentonite addition on heavy metal mobility; and (3) to provide suggestions on the appropriate amount of bentonite for curbing heavy metal-contaminated soil.

## 2. Materials and Methods

### 2.1. Materials

The calcium bentonite samples were collected from the Xiangshan area, Zhongwei city, Ningxia Hui Autonomous region of China. All the samples were oven-dried at 120 °C, and then ground in an agate mortar and passed through a 200-mesh sieve. The chemical properties of bentonite samples are listed in Table 1.

**Table 1.** Chemical properties of bentonite.

| Components | $SiO_2$ | $Al_2O_3$ | $Fe_2O_3$ | MgO | CaO | $K_2O$ | $Na_2O$ | Loss on Ignition |
|---|---|---|---|---|---|---|---|---|
| Percentage (%) | 59.8 | 19.6 | 6.0 | 2.3 | 1.4 | 1.3 | 0.7 | 8.9 |

The contaminated soil samples were collected from the old tailings area of Jinchang city (38°32′ N, 102°08′ E; m.a.s.l.—1605 m). The tailings area covers an area of more than 6 km$^2$, where the main remaining heavy metals are Ni and Cu. The soil in the Ni and Cu tailings is mainly composed of fine silt and sand left after the ore was ground and cleaned. This area has undergone serious water and wind erosion; its soil has low water and nutrient-holding capacity, and its vegetation is in poor condition. Rising dust can cause serious air pollution and bring severe threats to human health [26]. Three sampling points were selected randomly, and soil samples (0–20 cm) were collected after the surface litter was carefully removed by hand. All the samples were air-dried and sieved through a 100-mesh sieve to remove the root and plant residues. The properties of polluted soil samples are listed in Table 2. Bentonite was evenly added into the 1.0 kg contaminated soil samples with mass ratios of 5%, 10%, 30%, 50%, and 70%, while no bentonite was added in the control group (CK). Three replications were arranged for each treatment. After being fully mixed, all samples were sprayed with deionized water to adjust the water content to 70% of the field water-holding capacity, and then were incubated at room temperature (15–25 °C) for 4 months.

**Table 2.** Properties of heavy metal contaminated soil.

| | Soil Particle Size Distribution (%) | | | | | Available Nutrients | | | | |
|---|---|---|---|---|---|---|---|---|---|---|
| Coarse (>0.25 mm) | Fine (0.25–0.05 mm) | Clay + Silt (<0.05 mm) | Bulk Density (g/cm$^3$) | pH | SOM (mg/kg) | AN (mg/kg) | AP (mg/kg) | AK (mg/kg) | Total Cu (mg/kg) | Total Ni (mg/kg) |
| 44.5 | 23.7 | 31.8 | 1.2 | 7.2 | 0.9 | 2.4 | 0.6 | 12.4 | 3122 | 1886 |

Soil physicochemical properties and heavy metal form were determined through a five-point sampling method. Soil pH, soil organic matter (SOM), available nitrogen (AN), available phosphorus (AP), and available potassium (AK) were analyzed with standard methods issued by the Chinese Ecosystem Research Network [27].

## 2.2. Separation and Measurements of Heavy Metal Fractions

Soil samples were taken from each treatment, and then air-dried, ground, and sieved with a 60-mesh sieve. One gram of sieved soil was put into a 50 mL centrifuge tube, and the heavy metal fractions were separated by the sequential extraction method [12].

(1) Exchangeable: A total of 16 mL of $MgCl_2$ solution with a concentration of 1 mol/L and a pH of 7.0 was placed in the above-mentioned centrifuge tubes. The tubes were continuously agitated at 25 °C for 1 h, and the solutions were then centrifuged for 25 min. The samples were left standing, and the supernatant was subsequently taken and diluted in a 50 mL volumetric flask for further measurements. The residue was washed with distilled water.

(2) Bound to carbonates: The residue from step (1) was leached with 16 mL of 1 mol/L NaAc and the pH of the solution was adjusted to 5.0. The tube was shaken gently at 25 °C for 8 h, and continuous agitation was maintained for 25 min. The samples were then left standing and the supernatant was taken and diluted in a 50 mL volumetric flask for further measurements. The residue was washed with distilled water, and then the solution was centrifuged and the supernatant was removed.

(3) Bound to Fe-Mn oxides: The residue from step (2) was extracted with 16 mL of 0.04 mol/L $NH_2OH{\bullet}HCl$ solution in 25% (*v/v*) of HOAc. The tube was occasionally agitated at 96 °C for 4 h, and then it was centrifuged for 25 min. The samples were left standing and the supernatant was taken and diluted in a 50 mL volumetric flask for further measurements. The residue was washed with distilled water, and then the solution was centrifuged and the supernatant was removed.

(4) Bound to organic matter: A total of 3 mL of 0.01 mol/L $HNO_3$ solution and 5 mL of 30% $H_2O_2$ solution adjusted to a pH of 2 with $HNO_3$ were added to the centrifuge tube containing the residue of step (3), and the mixture was heated to 85 °C in a water-bath and then agitated for 2 h occasionally. A second 5 mL of 30% $H_2O_2$ solution adjusted to a pH of 2 was then added. The mixture was heated at 85 °C for 2 h with intermittent agitation. After the samples were cooled to room temperature, 5 mL of 3.2 mol/L $NH_4Ac$ solution in 20% (*v/v*) of $HNO_3$ was added and they were diluted to 20 mL and agitated continuously for 35 min, after which the tubes were centrifuged for 25 min. The samples were left standing and the supernatant was taken and diluted in a 50 mL volumetric flask for further measurements. The residue was washed with distilled water, and then the solution was centrifuged and the supernatant was removed.

(5) Residual: The residue from step (4) was digested with $HF$-$HClO_4$-$HNO_3$ mixture (1 mL of HCl, 3 mL of $HNO_3$, 1 mL of HF, and 1 mL of $H_2O_2$), and then the contents of Ni and Cu in solution were determined by atomic absorption spectrometry.

## 2.3. Statistical Analysis

The mobility index (MI) was used to evaluate the mobility of heavy metals in soil [28]; the calculation equation is as follows:

$$MI = \sum_{i=1}^{n} \frac{Fi/Ti}{n} \tag{1}$$

where Fi is the concentration of exchangeable-form heavy metals, Ti is the total concentration, and n is the number of soil samples.

Statistical software SPSS 13.0 for Windows (SPSS Inc. Chicago, IL, USA) was used to test the significance of the data.

## 3. Results

### 3.1. Effects of Bentonite on Soil Physicochemical Properties

Table 3 shows that soil pH increased with an increasing level of bentonite addition, ranging from 7.2 to 9.1. SOM, AN and AP were reduced by the addition of bentonite, with decreases ranging from 2% to 19%, 5% to 17%, and 7% to 33%, respectively. The addition of bentonite enhanced the concentration of AP by 1–10% (Table 3).

**Table 3.** Soil pH and the contents of available nutrients under different treatments.

| Treatment | pH | SOM % | AN mg/kg | AP mg/kg | AK mg/kg |
|---|---|---|---|---|---|
| CK | 7.2 | 0.9 | 2.4 | 0.6 | 12.4 |
| 5% | 7.3 | 0.8 | 2.0 | 0.5 | 12.7 |
| 10% | 7.9 | 0.8 | 1.9 | 0.5 | 12.9 |
| 30% | 8.1 | 0.7 | 1.8 | 0.4 | 13.2 |
| 50% | 9.0 | 0.6 | 1.5 | 0.3 | 14.4 |
| 70% | 9.1 | 0.5 | 1.4 | 0.2 | 15.8 |

Note: SOM is soil organic matter; AN is available nitrogen; AP is available phosphorus; AK is available potassium.

### 3.2. Effects of Bentonite on the Form of Heavy Metals in Soil

The residual was the main fraction of Ni and Cu, accounting for 53% and 57% of the total amount in the control group of soil, respectively, followed by that of bound to organic matter (20% and 16%), bound to Fe-Mn oxides (18% and 12%), bound to carbonates (6% and 7%) and exchangeable (3% and 4%) (Tables 4 and 5). The content of each fraction was in the order: Residual > bound to organic matter > bound to Fe-Mn oxides > bound to carbonates > exchangeable.

**Table 4.** Chemical forms of Ni in soil treated with different amounts of bentonite (mg/kg).

| Treatment | Total Amount | Exchangeable | Bound to Carbonates | Bound to Fe-Mn Oxides | Bound to Organic Matter | Residual | Absorptivity (%) |
|---|---|---|---|---|---|---|---|
| CK | 480 | 16 | 28 | 85 | 95 | 252 | 25 |
| 5% | 628 | 9 | 17 | 97 | 106 | 378 | 32 |
| 10% | 911 | 7 | 13 | 147 | 161 | 577 | 47 |
| 30% | 1305 | 3 | 4 | 209 | 229 | 824 | 66 |
| 50% | 1683 | 1 | 2 | 271 | 297 | 1068 | 86 |
| 70% | 1818 | 1 | 1 | 289 | 320 | 1080 | 89 |

**Table 5.** Chemical forms of Cu in soil treated with different amounts of bentonite (mg/kg).

| Treatment | Total Amount | Exchangeable | Bound to Carbonates | Bound to Fe-Mn Oxides | Bound to Organic Matter | Residual | Absorptivity (%) |
|---|---|---|---|---|---|---|---|
| CK | 1067 | 41 | 68 | 109 | 151 | 576 | 30 |
| 5% | 1089 | 29 | 35 | 133 | 169 | 706 | 34 |
| 10% | 1574 | 15 | 19 | 207 | 253 | 1057 | 49 |
| 30% | 2141 | 8 | 7 | 251 | 346 | 1492 | 66 |
| 50% | 2893 | 2 | 3 | 301 | 448 | 2052 | 89 |
| 70% | 2995 | 1 | 1 | 307 | 418 | 2104 | 92 |

After the addition of bentonite, the content of exchangeable and bound to carbonates fraction decreased, but the content of bound to Fe-Mn oxides, bound to organic matter and residual increased. For example, with 5% of bentonite addition, the fraction of exchangeable fraction and bound to carbonates decreased from 16 and 28 mg/kg to 9 and 17 mg/kg for Ni, and the content bound to Fe-Mn oxides, bound to organic matter, and residual increased from 85, 95 and 252 mg/kg to 97, 106 and 378 mg/kg, respectively. Other treatments have the same tendency. The most significant changes occurred in 50% of bentonite addition,

while their absorptivity could reach to 86% to 89% for Ni and Cu, respectively. In addition, the maximum absorptivity could reach 89% and 92% with 70% of bentonite addition.

The proportion of each fraction did not change with the increasing addition of bentonite. As shown in Figure 1, the exchangeable fraction of Ni and Cu significantly decreased following the addition of bentonite. The proportion of the fraction bound to Fe-Mn oxides, bound to organic matter, and residual considerably increased with the addition of bentonite, ranging from 16% to 65% for Ni and 11% to 74% for Cu, respectively. The proportion of the bound to carbonates fraction slightly decreased with bentonite addition, ranging from 0 to 2% for Ni and from 0 to 3% for Cu.

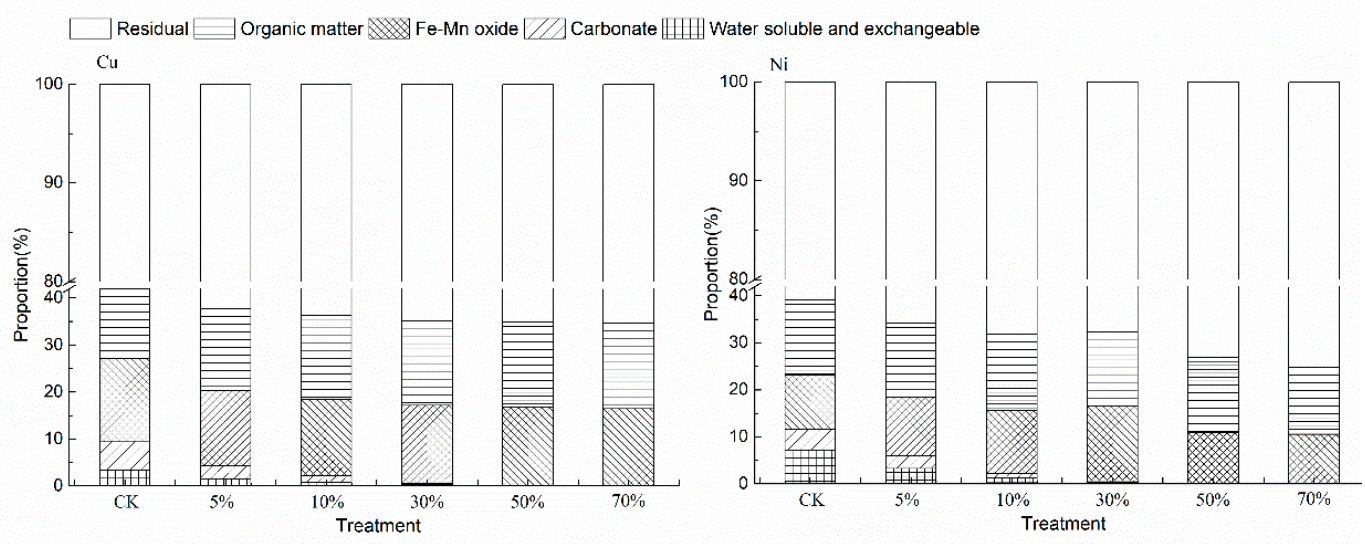

**Figure 1.** The proportion of each form of Ni and Cu in soil.

### 3.3. Effects of Bentonite on Mobility of Heavy Metals in Soil

The mobility of heavy metals in soil is characterized by the mobility index (MI). As shown in Figure 2, the MI of Ni and Cu decreased with an increasing level of bentonite; it was reduced by 9% and 20% with 5% of bentonite addition, respectively, but decreased from 0.35 and 0.21 in CK to 0.11 and 0.09 in the treatment with 70% bentonite addition, i.e., decreases of 68% and 56%, respectively.

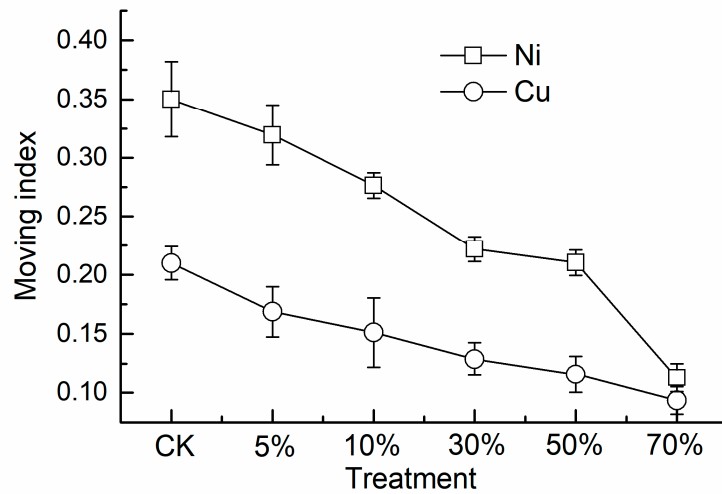

**Figure 2.** Changes in mobility index of Ni and Cu after the addition of bentonite.

## 4. Discussion

Our findings indicated that bentonite addition plays a significant role in the treatment of soils that have been subjected to heavy metal pollution. In this study, the pH value and the content of AP increased, but the contents of SOM, AN, and AP decreased with the addition of bentonite. The main reason for this is that, as a porous material with a large specific surface area, bentonite has strong adsorption [24]. A large number of studies showed that the addition of such adsorbents can enhance the adsorption of alkali-hydrolyzed nitrogen, AP, and AK in soil [10,29]. In this experiment, although bentonite enhanced the adsorption of soil for AP, it increased the amount of potassium and pH value, which ultimately resulted in an increase in AK content. This is consistent with the study of Lin et al. [24]. Thus, the addition of bentonite could regulate the release of available nutrients and prolong their effective supply time in the soil [30]. Nevertheless, the amount of bentonite needs to be controlled in a certain range, and excessive application may lead to plant malnutrition [31].

After bentonite was added, the physicochemical properties such as pH value and organic matter of soil changed, affecting the form of heavy metals and further influencing their bioavailability [32]. The soil pH value not only affects the ion composition of soil solution and various chemical reactions in soil, but also significantly modifies the forms of heavy metals and their absorption by plants, which is an important factor in determining the remediation effects for soil contaminated by heavy metals [33]. In this study, from Tables 3–5, we can see that the content of exchangeable heavy metals was negatively correlated with pH value. The main reason for this is that the negative charges on the surface of clay minerals, hydrated oxides, and organic matter increased with the rising pH value, which led to the enhancement of the adsorption capacity of heavy metals and the reduction in ion concentration of heavy metals in soil solution [30]. Secondly, the specific adsorption of heavy metals such as Ni and Cu on the surface of oxides increased with the increasing pH value, and most of the adsorbed heavy metals were performed so with a rising pH value [11]. Thirdly, the stability of soil–organic matter–metal complex increased with the pH value, which resulted in a decrease in the concentration of heavy metals in the solution [21]. Fourthly, with the increase in pH value, the concentration of the ions of iron, aluminum, and magnesium in the soil solution decreased, facilitating the adsorption of heavy metal ions by soil [24]. When the pH value is low enough, the release of heavy metals due to the dissolution of carbonates and their absorption by plants can make the heavy metal form transferable between the exchangeable form and the bound carbonates form [34]. The change in organic matter induced by the addition of bentonite is another reason for the change in heavy metal form. It has been reported that the content of the carbonate-bound form was negatively correlated with organic matter content. The contents of exchangeable and organic matter bound fractions were directly proportional to the content of organic matter, and the addition of organic matter may facilitate the transformation of carbonate bound to organic matter bound fraction [35]. The possible reason behind this is that a large number of functional groups in organic matter have a strong adsorption capacity for heavy metal ions, and the humic acid produced through humus decomposition could form complexes with heavy metals in soil, thus increasing the content of the organic matter bound fraction. Meanwhile, the organic matter was conducive to the activation of iron oxides, so that the content of the heavy metal fraction bound to soil oxides was directly proportional to the content of organic matter [36]. The increase in organic matter concentration in the soil can reduce the content of soluble heavy metals. For example, the content of available cadmium in soil decreased significantly, by approximately 40%, following the application of organic fertilizer [37]. It was found that soluble heavy metals in yellow clay soil in northwestern China decreased by 60% to 80% after humic acid addition, while the fraction bound to carbonates and organic matter increased [38]. As an effective adsorbent, organic matter can greatly reduce the activity of ions of heavy metals, thus resulting in a decrease in the contents of soluble heavy metals in soil [39]. In addition,

the content of rare earth elements, water, and dolomite and the particle size distribution has an influence on the composition of different forms of heavy metals [6].

There are various fractions of heavy metals in soil. Our findings indicated that residual was the main fraction of both Ni and Cu, accounting for 64% and 73%, respectively, which suggested high stability of heavy metals in soil. The proportion of the bound to organic matter fraction and the exchangeable fraction of Cu was greater than that of Ni, while the proportion of the residual fraction was significantly less than that of the latter, which is related to the strong binding ability between Cu and organic matter. Cu can form metal ions of internal coordination compounds with two or more organic functional groups (mainly carboxyl, hydroxyl, and phenol) in soil organic matter, and bivalent Cu in the environment can complex with some components in soil [39]. Various fractions of heavy metals are often in a dynamic equilibrium state. The soluble form was mainly supplemented by the exchangeable fraction when they were reduced due to the absorption by plants, while the increase in absorbable fractions due to the external input promoted the transformation from the exchangeable to residual fraction, which is difficult to absorb. There was a certain equilibrium state among these fractions for some time, which changed constantly with the variations of environments (such as plant absorption, integration, temperature, and water regimes), thus achieving remediation [6,13]. Our results showed that the mobility index (MI) of Ni and Cu considerably decreased with the addition of bentonite, indicating the reduction in the potential threat of heavy metals to the environment, which was mainly attributable to the decrease in the contents of the exchangeable fractions of these metals. Therefore, the addition of bentonite in heavy metal-contaminated soil can improve soil quality; meanwhile, we suggest that the optimal ratio of bentonite should be selected in a heavy metal-contaminated soil.

## 5. Conclusions

Bentonite addition can significantly affect the speciation and mobility of Cu and Ni in soils from old mine tailings. The content of exchangeable and bound to carbonates fraction decreased, but the content of bound to Fe-Mn oxides, bound to organic matter and residual increased with bentonite addition and the most significant changes occurred in 50% of bentonite addition. However, the proportion of each fraction did not vary much with the addition of bentonite. The proportion of the exchangeable fractions of Ni and Cu in the soil significantly decreased due to the addition of bentonite, while there was an increase in the bound to Fe-Mn fraction, bound to organic matter fraction, and residual fraction. The addition of bentonite can significantly reduce the mobility index of Ni and Cu in the soil, which means that the addition of bentonite can effectively fix the heavy metals in the soil.

**Author Contributions:** Data curation, Y.G. and X.L.; resources, Y.G. All authors have read and agreed to the published version of the manuscript.

**Funding:** This research received no external funding.

**Institutional Review Board Statement:** Not applicable.

**Informed Consent Statement:** Not applicable.

**Data Availability Statement:** The data used to support the findings of this study are available from the corresponding author upon reasonable request.

**Acknowledgments:** The authors gratefully acknowledge the editor and several anonymous reviewers for valuable comments on the manuscript. Thanks also are given to the staff at the Shapotou Desert Research and Experiment Station for their assistance.

**Conflicts of Interest:** The authors declare no conflict of interest.

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
