# Peer review of "Effects of Bentonite Addition on the Speciation and Mobility of Cu and Ni in Soils from Old Mine Tailings"

_sustainability, doi:10.3390/su141710878_

Round 1

Reviewer 1 Report

This manuscript presents results of an experiment where bentonite has been added into a soil contaminated with Cu and Ni. The distribution of these metals into fractions bound by different soil constituents has been determined using a well-known Tessier et al. procedure. The topic is important and within the scope of this journal.

The basic fundamental reason for this negative recommendation are the results themselves. The authors report that addition of bentonite decreases the total concentration of Cu and Ni (even down to one third of the original) in the soil. As far as I can understand, this outcome is impossible. The total concentration, calculated as the sum of the different fractions, involves a digestion with HF-HClO4-HNO3 as the final step, and this strong dissolution should dissolve all forms of Cu and Ni, even the most sparingly soluble ones. These metals cannot escape from the system as gaseous losses, and, as this was an incubation experiment, leaching out of the system did not take place. This is why the total amount (sum of fractions) should remain the same in all treatments.

A possible explanation of the decrease of the total concentrations of Cu and Ni may be a dilution effect. In the Materials & methods section (L 111-112) it is said that ”bentonite was evenly added into the contaminated soil samples with the mass ratio of 5%, 10%, 30%, 50% and 70%” but in Tables 3-5 and Figures 1-2, the additions are labelled as 1, 2, 3, 5 and 7%. This is entirely confusing, and the authors need to clear out this confusion.

There are also several other issues that need to be revised but they are not as fundamental as the issue above. I present a rough list of criticism as follows:

A revised title is proposed: Effects of bentonite on the forms of Cu and Ni in soil

Abstract: What is the rationale of adding bentonite into the soil to alleviate heavy metal contamination? A brief statement is needed. – Here (L15) the decrease of total content is mentioned. It is just confusing to the reader and it should be explained in a reasonable manner. – Accuracy of numerical results: The distribution and changes of different forms of Cu and Ni should be presented as full percentages. Using of decimals is unrealistic accuracy. This same remark applies to the whole paper.

Introduction

L35: What does ”III level” stand for?

L43-63: This section presents the well-established fact that total concentration of an element seldom is a good indicator for bioavailability but different species need to be determined. This elementary concept could be presented in a brief manner. The readers of this paper are familiar with this issue.

L71: Tell something more about bentonite. What is it mineralogically? Where does it occur?  

L73: iron -> ion

L76: Changes of pH should NEVER be presented as percent changes.

L87: testify -> test   

Materials and methods:

L96 and many other places: Give the sieving characteristics also in millimeters, additional to mesh, because the mesh is not known by readers anymore.

Table 1: Use only one decimal

L104: ”low capacity of water and nutrient conservation” What does it mean? Do you mean low water and nutrient holding capacity?

L111-112: The mass ratio of 5%, 10%, 30%, 50% and 70% does not correspond to the bentonite additions mentioned in Tables and Figures of this paper. Please clear this contradiction out.

L118: You have to briefly describe the methods also here. It is not sufficient to refer to another paper only.

Table 2: Particle size distribution should be presented with full percentages. – The bulk density of 3.16 g/cm3 is an unrealistic value. It needs clarification. Usually, soil mineral particles (i.e., solid particles, without pore space) have a particle density of 2.65 g/cm3. Is the high ”bulk density” due to the heavy metals? For example, the density of pyrite is about 5 g/cm3. Are you presenting bulk density or particle density? – What are the units of SOM, AN, AP and AK? – What fraction do Cu and Ni stand for? Total? These results should be presented without decimals.

L125: continuous -> sequential

L140: What does ”at 96” stand for?

L157: What is the rationale of adding 1 drop of concentrated HNO3 into HF-HClO4-HNO3 extracts for conservation?

How was total Cu and Ni concentrations determined? If they were determined by the summation of fractions, this should be clearly mentioned here.

Results:

L171: Changes of pH should NEVER be presented as percentages.

L172: The highest pH was measured in 7% addition, not at 5% addition.

L187: ”both the total amount and the content of each component decreased”. Please see my comment at the beginning. This is most confusing and should be cleared out.

Table 4 and 5: All results should be presented as full mg/kg, without decimals. The decimals are unrealistic.

Fig. 1. It would be better to present all the columns for Cu together and all columns of Ni together.

Discussion:

There are many statements that are against what is presented in the tables. For example, on L259, it is said that ”the content of heavy metal fraction bound to carbonates was positively correlated with pH”. This is not the case on the basis of the results presented in Tables 4 and 5. The same applies to the statement about organic matter bound forms on L264.

On L273, the sentence deals with the change of organic matter induced by the addition of bentonite. This statement is nonsense to me. What mechanism is there that could decrease soil organic matter upon addition of bentonite?

The Discussion contains also many other questionable statements, e.g., L283-285 but I leave them to the authors to clear out.

L315: How did bentonite provide nutrients to plants? The authors reported that available N and P were decreased upon bentonite addition.

No comparison was made to other studies where the Tessier fractionation scheme has been used.

The authors may want to add some discussion about the mobility of the metals in their soils. The original pH of their soil was 7.15. At this high pH heavy metals are hardly mobile, and further elevation of the pH may not much help in the decrease of mobility. Is it the adsorption capacity that is brought in by the bentonite?

Conclusions:

This section should not be just repetition of the main findings, nor should it be repetition of the Abstract. What did you actually find out? What is your message? Those are the essential contents of the Conslusions section.

References: This study has 45 references. It is an adequate number. However, only 9 out of the 45 references are from outside China. This kind of studies have been carried out outside of China and those studies should be adequately referenced.

The English language is not adequate, and the proof-reading has not been properly carried out.

Author Response

This manuscript presents results of an experiment where bentonite has been added into a soil contaminated with Cu and Ni. The distribution of these metals into fractions bound by different soil constituents has been determined using a well-known Tessier et al. procedure. The topic is important and within the scope of this journal.

The basic fundamental reason for this negative recommendation are the results themselves. The authors report that addition of bentonite decreases the total concentration of Cu and Ni (even down to one third of the original) in the soil. As far as I can understand, this outcome is impossible. The total concentration, calculated as the sum of the different fractions, involves a digestion with HF-HClO4-HNO3 as the final step, and this strong dissolution should dissolve all forms of Cu and Ni, even the most sparingly soluble ones. These metals cannot escape from the system as gaseous losses, and, as this was an incubation experiment, leaching out of the system did not take place. This is why the total amount (sum of fractions) should remain the same in all treatments.

Re: The main mineral component of bentonite is montmorillonite, which has a certain repair effect on various heavy metal elements. Studies have shown that bentonite can be used for the remediation of heavy metal soilpollution. Bentonite and heavy metals will generate Pb(NO3)2 and CdCO3 and other substances, so that the precipitation rate of heavy metals is above 85%. Other studies have shown that the application of bentonite and the combined treatment of bentonite and arbuscularmycorrhizal fungi can effectively reduce the content of available heavy metals in the soil, and the content of heavy metals in the above-ground and underground parts of the plant is also significantly reduced. Xie et al. used bentonite to remediate lead and zinc polluted soil in mining areas. The study found that under the condition of pH=5, when bentonite:contaminated soil=1:5, the remediation effect is the best. Another researcher loaded chitosan on bentonite to make a granular adsorbent for the remediation of zinc-contaminated soil. The results showed that the mass ratio of bentonite and chitosan was 1:1 under the condition of pH=6-8, the removal rate of heavy metal zinc reaches 98%.Naseemet al showed that when treated Pb2+ with purified bentonite, the removal rate could reach 95%.

A possible explanation of the decrease of the total concentrations of Cu and Ni may be a dilution effect. In the Materials & methods section (L 111-112) it is said that ”bentonite was evenly added into the contaminated soil samples with the mass ratio of 5%, 10%, 30%, 50% and 70%” but in Tables 3-5 and Figures 1-2, the additions are labelled as 1, 2, 3, 5 and 7%. This is entirely confusing, and the authors need to clear out this confusion.

RE: sorry, we made a mistake here, the table and figure labels should be 5%, 10%, 30%, 50% and 70%, we have changed in our revised manuscript.

There are also several other issues that need to be revised but they are not as fundamental as the issue above. I present a rough list of criticism as follows:

A revised title is proposed: Effects of bentonite on the forms of Cu and Ni in soil

RE: done. We have changed in our revised manuscript.

Abstract: What is the rationale of adding bentonite into the soil to alleviate heavy metal contamination? A brief statement is needed. – Here (L15) the decrease of total content is mentioned. It is just confusing to the reader and it should be explained in a reasonable manner. – Accuracy of numerical results: The distribution and changes of different forms of Cu and Ni should be presented as full percentages. Using of decimals is unrealistic accuracy. This same remark applies to the whole paper.

RE: done. We have changed in our revised manuscript.

Introduction

L35: What does ”III level” stand for?

RE: Environmental quality standard for soils, in this standard, the Ni and Cu in soil was 200 mg/kg and 400 mg/kg.

L43-63: This section presents the well-established fact that total concentration of an element seldom is a good indicator for bioavailability but different species need to be determined. This elementary concept could be presented in a brief manner. The readers of this paper are familiar with this issue.

RE: done. We have changed in our revised manuscript.

L71: Tell something more about bentonite. What is it mineralogically? Where does it occur?  

RE: done. We have changed in our revised manuscript.

L73: iron -> ion

RE: done. We have changed in our revised manuscript.

L76: Changes of pH should NEVER be presented as percent changes.

RE: done. We have changed in our revised manuscript.

L87: testify -> test   

RE: done. We have changed in our revised manuscript.

Materials and methods:

L96 and many other places: Give the sieving characteristics also in millimeters, additional to mesh, because the mesh is not known by readers anymore.

RE: done. We have changed in our revised manuscript.

Table 1: Use only one decimal

RE: done. We have changed in our revised manuscript.

L104: ”low capacity of water and nutrient conservation” What does it mean? Do you mean low water and nutrient holding capacity?

RE: yes. We have changed in our revised manuscript.

L111-112: The mass ratio of 5%, 10%, 30%, 50% and 70% does not correspond to the bentonite additions mentioned in Tables and Figures of this paper. Please clear this contradiction out.

RE: sorry, we made a mistake here, the table and figure labels should be 5%, 10%, 30%, 50% and 70%, we have changed in our revised manuscript.

L118: You have to briefly describe the methods also here. It is not sufficient to refer to another paper only.

RE: done. We have changed in our revised manuscript.

Table 2: Particle size distribution should be presented with full percentages. – The bulk density of 3.16 g/cm3 is an unrealistic value. It needs clarification. Usually, soil mineral particles (i.e., solid particles, without pore space) have a particle density of 2.65 g/cm3. Is the high ”bulk density” due to the heavy metals? For example, the density of pyrite is about 5 g/cm3. Are you presenting bulk density or particle density? – What are the units of SOM, AN, AP and AK? – What fraction do Cu and Ni stand for? Total? These results should be presented without decimals.sorry, it is 1.16 here,

RE: we have checked our data and changed in our revised manuscript.

L125: continuous -> sequential

RE: done. We have changed in our revised manuscript.

L140: What does ”at 96” stand for?

RE: it was 96℃. We have changed in our revised manuscript.

L157: What is the rationale of adding 1 drop of concentrated HNO3 into HF-HClO4-HNO3 extracts for conservation?How was total Cu and Ni concentrations determined? If they were determined by the summation of fractions, this should be clearly mentioned here.

RE:The total amount of heavy metals and the content of heavy metals in the residue were digested by triacid (HF-HNO3-HClO4) method and determined by atomic absorption spectrophotometry (GBC 932AA, Australia).

Results:

L171: Changes of pH should NEVER be presented as percentages.

RE: done. We have changed in our revised manuscript.

L172: The highest pH was measured in 7% addition, not at 5% addition.

RE: done. We have changed in our revised manuscript.

L187: ”both the total amount and the content of each component decreased”. Please see my comment at the beginning. This is most confusing and should be cleared out.

RE: done. We have changed in our revised manuscript.

Table 4 and 5: All results should be presented as full mg/kg, without decimals. The decimals are unrealistic.

RE: done. We have changed in our revised manuscript.

Fig. 1. It would be better to present all the columns for Cu together and all columns of Ni together.

RE: done. We have changed in our revised manuscript.

Discussion:

There are many statements that are against what is presented in the tables. For example, on L259, it is said that ”the content of heavy metal fraction bound to carbonates was positively correlated with pH”. This is not the case on the basis of the results presented in Tables 4 and 5.

RE: done. We have changed in our revised manuscript.

The same applies to the statement about organic matter bound forms on L264.

RE: done. We have changed in our revised manuscript.

On L273, the sentence deals with the change of organic matter induced by the addition of bentonite. This statement is nonsense to me. What mechanism is there that could decrease soil organic matter upon addition of bentonite?

RE: The change of organic matter caused by the addition of bentonite is also one of the reasons for the change of heavy metal forms. Some studies have shown that carbonate-bound heavy metals are negatively correlated with the content of organic matter, but the correlation is not significant. Exchange and organically bound heavy metals are positively correlated with organic matter, increasing the content of organic matter can convert the carbonate-bound state to the organic-bound state. The mechanism may be that organic matter has a large number of functional groups, which has a strong adsorption capacity for heavy metal ions, and the decomposition of humus forms a complex that can be formed by humic acid and heavy metals in the soil, thereby increasing the content of organic heavy metals. At the same time, the presence of organic matter is conducive to the activation of iron oxide, so that the content of heavy metals in soil oxide-bound state is proportional to the content of organic matter.

The Discussion contains also many other questionable statements, e.g., L283-285 but I leave them to the authors to clear out.

RE:The decrease in the content of soluble heavy metals is due to the fact that most of the organic matter is an effective adsorbent, which can greatly reduce the activity of heavy metal ions, thereby reducing the content of soluble heavy metals in the soil. However, some other studies also have different conclusions.

L315: How did bentonite provide nutrients to plants? The authors reported that available N and P were decreased upon bentonite addition.

RE: done. We have changed in our revised manuscript.

No comparison was made to other studies where the Tessier fractionation scheme has been used.

RE: done. We have added in our revised manuscript.

The authors may want to add some discussion about the mobility of the metals in their soils. The original pH of their soil was 7.15. At this high pH heavy metals are hardly mobile, and further elevation of the pH may not much help in the decrease of mobility. Is it the adsorption capacity that is brought in by the bentonite?

RE:It may be thepolluted soils with heavy metals, so the pH is a little higher.

Conclusions:

This section should not be just repetition of the main findings, nor should it be repetition of the Abstract. What did you actually find out? What is your message? Those are the essential contents of the Conslusions section.

RE: done. We have changed in our revised manuscript.

References: This study has 45 references. It is an adequate number. However, only 9 out of the 45 references are from outside China. This kind of studies have been carried out outside of China and those studies should be adequately referenced.

RE: done. We have added in our revised manuscript.

The English language is not adequate, and the proof-reading has not been properly carried out.

RE: Thanks very much for your comments. We have asked Prof. Xiaojun Li and Yafeng Zhang, they are well established experts, to polish our paper. Please see if the revised version met the English presentation standard.

Reviewer 2 Report

This manuscript studied the effect of bentonite on heavy metals Cu and Ni in soil. The detailed comments are as follows:

1.     Line44-45Please seriously consider the accuracy of this expression. In my opinion, plants have an important impact on the content and form of heavy metals in soil, and there are many studies on phytoremediation of heavy metal pollution. In addition, the following referenceTherefore, to evaluate the mobility, bioavailability and toxicity of heavy metals in soil, not only the total content but also the forms of them should be analyzed. This clearly does not constitute a causal relationship.

2.     Line71 Please compare the advantages and disadvantages of existing soil amendments and, if possible, list the tables.

3.     Table2 What do SOM, AN, AP mean? Although line116-line119 has been explained, the expression needs to be improved. For example, the expression should be changed to organic matter (SOM), available nitrogen (AN).

4.     Line170 Please check the accuracy of the data. The pH range in Table 3 is 7.15 - 9.08, which is inconsistent with the description.

5.     Section3.1 With the increase of bentonite content, the weight percentage of AN, AP and AK will inevitably decrease. I mean, if quartz sand is added instead of bentonite AN, AP, AK, etc., the mass percentage will still decrease, how can we determine that it is completely affected by the addition of bentonite? In my opinion, the author should compare the difference between the actual value and the theoretical value to determine the impact of bentonite.

6.     There is a serious problem of data mismatch in this manuscript, line 111 ''the mass ratio of 5 %, 10 %, 30 %, 50 % 111 and 70 % '', while treatment in Table 3 4 and 5 is 1 %, 2 %, 3 %, 5 % and 7 %.

7.     Please supplement the units in Tables 4 and 5.

8.     Line233 There seems to be no more data to support this argument.

9.     Line 243-244This description is inconsistent with the data. It can be seen from the data that exchangeable heavy and carbonates bound increase with the increase of bentonite addition, that is, increase with the increase of pH. Obviously, the author refers to the percentage change of their element forms, so the author needs to add relevant charts in the text.

10.  The concluding part should be rewritten. The conclusion is the summary and improvement of the article, which should include the combination of research, rules and new findings, rather than just listing the relevant data of the experiment.

11.  Where is the innovation of this paper? Only considering the influence of bentonite on the heavy metal forms of Cu and Ni, which is not enough for research articles.

In conclusion, the authors should carefully examine the experimental results and related data expressions to ensure the rigor of the article. In addition, the conclusion should be rewritten. For these reasons, I recommend revision.

Author Response

This manuscript studied the effect of bentonite on heavy metals Cu and Ni in soil. The detailed comments are as follows:

  1. Line44-45,Please seriously consider the accuracy of this expression. In my opinion, plants have an important impact on the content and form of heavy metals in soil, and there are many studies on phytoremediation of heavy metal pollution. In addition, the following reference“Therefore, to evaluate the mobility, bioavailability and toxicity of heavy metals in soil, not only the total content but also the forms of them should be analyzed.” This clearly does not constitute a causal relationship.

RE: we have changed in our revised manuscript.

  1. Line71 Please compare the advantages and disadvantages of existing soil amendments and, if possible, list the tables.

RE: we have changed in our revised manuscript.

  1. Table2 What do SOM, AN, AP mean? Although line116-line119 has been explained, the expression needs to be improved. For example, the expression should be changed to organic matter (SOM), available nitrogen (AN).

RE: we have changed in our revised manuscript.

  1. Line170 Please check the accuracy of the data. The pH range in Table 3 is 7.15 - 9.08, which is inconsistent with the description.

RE: we have changed in our revised manuscript.

  1. Section3.1 With the increase of bentonite content, the weight percentage of AN, AP and AK will inevitably decrease. I mean, if quartz sand is added instead of bentonite AN, AP, AK, etc., the mass percentage will still decrease, how can we determine that it is completely affected by the addition of bentonite? In my opinion, the author should compare the difference between the actual value and the theoretical value to determine the impact of bentonite.

RE:this manuscript only discusses the effects of bentonite addition on the speciation of heavy metals Cu and Ni in soil. In our other manuscript, we have studied theadsorption characteristics of bentoniteaddition on Ni and Cu. We found that withincreased addition of bentonite amount, the concentration of Ni2+ and Cu2+ gradually decreased. When the ratio of bentonite to contaminated soil is 4:1, the concentration in balanced solution approached to 0.

  1. There is a serious problem of data mismatch in this manuscript, line 111 ''the mass ratio of 5 %, 10 %, 30 %, 50 % 111 and 70 % '', while treatment in Table 3 4 and 5 is 1 %, 2 %, 3 %, 5 % and 7 %.

RE:sorry, we made a mistake here, the table and figure labels should be 5%, 10%, 30%, 50% and 70%, we have changed in our revised manuscript.

  1. Please supplement the units in Tables 4 and 5.

RE: we have added in our revised manuscript.

  1. Line233 There seems to be no more data to support this argument.

RE: we have changed in our revised manuscript.

  1. Line 243-244,This description is inconsistent with the data. It can be seen from the data that exchangeable heavy and carbonates bound increase with the increase of bentonite addition, that is, increase with the increase of pH. Obviously, the author refers to the percentage change of their element forms, so the author needs to add relevant charts in the text.

RE: we have changed in our revised manuscript.

  1. The concluding part should be rewritten. The conclusion is the summary and improvement of the article, which should include the combination of research, rules and new findings, rather than just listing the relevant data of the experiment.

RE: we have changed in our revised manuscript.

  1. Where is the innovation of this paper? Only considering the influence of bentonite on the heavy metal forms of Cu and Ni, which is not enough for research articles.

RE: we have changed in our revised manuscript.

In conclusion, the authors should carefully examine the experimental results and related data expressions to ensure the rigor of the article. In addition, the conclusion should be rewritten. For these reasons, I recommend revision.

RE: we have changed in our revised manuscript.

Reviewer 3 Report

Dear authors,

I suggest to change the title 'Effects of Bentonite on the form of heavy metals Cu and Ni in the soil' to 'Effects of bentonite addition on the speciation of heavy metals Cu and Ni in soil from an old tailings area of the Jinchang city (China)'.

Abstract should be revised so that it features a mini display of the manuscript (MS); the problem, methods, crucial results, short interpretation, and conclusion sentence. English is quite good, but a professional language service is required for the sake of better readability. Delete the first sentence as the heavy metal toxicity was not among the objectives of this experiment. L 8-9: total levels of Cu and Ni must be here so that it is clear that collected soil samples are heavily polluted. Here, the crucial detail is lacking: sequential extraction method! Provide a short description of your extraction method. L 13: two to max three significant figures are enough, and same holds for L 11 (e.g. 17.7 or 18%). L 17 instead of 'before and after', use 'prior to and following the...', it is more formal and appropriate for scientific article. L 17/18 this could be the 1st sentence of Abstract?

Introduction: L 32 (Liao et al...) this should be a new paragraph as you are talking about your study area, so make the transition logically – finish the previous (the beginning one) paragraph with soemthing which is related with ores, or metals, or similar. You mentioned mining, so give some citation(s), and provide few more relevant sentences. L 34 provide exact levels of total Cu and Ni in soil. L 47 'effectiveness', I am not hapy with the term; what do you mean by it? Please, change it. L 58 'easy'...? How? Please explain. L 61 why sediment when your work was done on soil? L 64 delete 'hot'. L 67 not alien, but allochtonous, or external. L 74-75 what do you mean by it? Is it good or bad for soil? L 79 does this mean that Cd should be also monitored while applying bentonite? L 88/89 how the second objective is different from the 1st one?

Metods: L 164 reference?

Results and discussion is correct. Still, try not to repeat in the text what has already been shown in Tables and figures. Always in the beginning (whatever the chapter or paragraph) emphasize the most important findings, i.e. declining levels of mobile Cu and Ni forms in treated soil, thus justifying efficiency of your experiment, as that was its main aim. 'Fourthly' is not correct term, delete it. All data should be presented with 2-3 significant figures.

Conclusion should be shorter, and again – focus on mobile Cu and Ni forms first, comparison with control samples, and then you can mention P, N, organic matter, pH, etc.

Author Response

I suggest to change the title 'Effects of Bentonite on the form of heavy metals Cu and Ni in the soil' to 'Effects of bentonite addition on the speciation of heavy metals Cu and Ni in soil from an old tailings area of the Jinchang city (China)'.

RE: done. We have changed in our revised manuscript.

Abstract should be revised so that it features a mini display of the manuscript (MS); the problem, methods, crucial results, short interpretation, and conclusion sentence.

RE: done. We have changed in our revised manuscript.

English is quite good, but a professional language service is required for the sake of better readability.

RE: Thanks very much for your comments. We have asked Prof. Xiaojun Li and Yafeng Zhang, they are well established experts, to polish our paper. Please see if the revised version met the English presentation standard.

Delete the first sentence as the heavy metal toxicity was not among the objectives of this experiment.

RE: done. We have changed in our revised manuscript.

L 8-9: total levels of Cu and Ni must be here so that it is clear that collected soil samples are heavily polluted. Here, the crucial detail is lacking: sequential extraction method! Provide a short description of your extraction method.

RE: done. We have changed in our revised manuscript.

L 13: two to max three significant figures are enough, and same holds for L 11 (e.g. 17.7 or 18%).

RE: done. We have changed in our revised manuscript.

L 17 instead of 'before and after', use 'prior to and following the...', it is more formal and appropriate for scientific article.

RE: done. We have changed in our revised manuscript.

L 17/18 this could be the 1st sentence of Abstract?

RE: done. We have changed in our revised manuscript.

Introduction: L 32 (Liao et al...) this should be a new paragraph as you are talking about your study area, so make the transition logically – finish the previous (the beginning one) paragraph with soemthing which is related with ores, or metals, or similar. You mentioned mining, so give some citation(s), and provide few more relevant sentences.

RE: done. We have changed in our revised manuscript.

L 34 provide exact levels of total Cu and Ni in soil.

RE: done. We have changed in our revised manuscript.

L 47 'effectiveness', I am not hapy with the term; what do you mean by it? Please, change it.

RE: done. We have changed in our revised manuscript.

L 58 'easy'...? How? Please explain.

RE:This refers to the migration between different forms of heavy metals in soil. The bound to organic and bound to Fe-Mn oxides have weak migration ability, which is called the potential effective state, but it is easier to transform into the effective state.

L 61 why sediment when your work was done on soil?

RE: here we mean theresidual was stable in soil, we have changed in our revised manuscript.

L 64 delete 'hot'.

RE: done. We have changed in our revised manuscript.

L 67 not alien, but allochtonous, or external.

RE: done. We have changed in our revised manuscript.

L 74-75 what do you mean by it? Is it good or bad for soil?

RE: sure, the addition of bentonite has positive effects on the treatment of heavy metals in soil.

L 79 does this mean that Cd should be also monitored while applying bentonite?

RE:This sentence just shows that bentonite also has adsorption effect on cadmiumL 88/89 how the second objective is different from the 1st one?

RE: the first one means heavy metalsforms,here is the heavy metal mobility, which means the mobility ofthe heavy metals in the soil

Metods: L 164 reference?

RE: done. We have added in our revised manuscript.

Results and discussion is correct. Still, try not to repeat in the text what has already been shown in Tables and figures. Always in the beginning (whatever the chapter or paragraph) emphasize the most important findings, i.e. declining levels of mobile Cu and Ni forms in treated soil, thus justifying efficiency of your experiment, as that was its main aim. 'Fourthly' is not correct term, delete it. All data should be presented with 2-3 significant figures.

RE: done. We have changed in our revised manuscript.

Conclusion should be shorter, and again – focus on mobile Cu and Ni forms first, comparison with control samples, and then you can mention P, N, organic matter, pH, etc.

RE: done. We have changed in our revised manuscript.

Reviewer 4 Report

Almost 70% of the literature is in Chinese language. Considerable part of the readers will be not able to verify the citations. In my opinion, this is not acceptable. The number of that kind of papers must be reduced to 8-10. Apart from that, some minor corrections should be made:

Line 46 - which mainly because …??? I think should be… mainly because

Line 61 – sediment ??? – I think should be… soil

Line 101, 126, 132, 133, 139, 144, 147-150, 158 – correct the upper and lower indices

Line 131 – the solution was centrifuged …time and g-force ???

Line 133 – should be …adjusted

Line 140 - at 96… ???

Line 148 – was heated at 86 …not 85℃ ???

Line 167 - SPSS 13.0 … please provide the manufacturer, city and country

Line 172, 320 - with the maximum appeared in the treatment with 5% of bentonite addition …not with 7% ???

Line 228 – adroption …should be adsorption

Line 243 - In this study, the content of exchangeable heavy metals was negatively correlated with pH value… correlated ??? I don’t see calculated correlations in this paper.

Line 253 - Fourthly, with the increase of pH value, the concentration of the ions of iron, aluminum and magnesium in the soil solution decreased, which facilitated the adsorption of heavy metal ions by soil. - How do you know that ??? I don’t see any results regarding the ions of iron, aluminum and magnesium in the soil solution. Missing citation ???

Line 256 - Finally, the product of multivalent cations and oxyhydrogen ions in soil solution increased with increasing pH value, which improved the chance of forming such heavy metal sediments.  - How do you know that ??? … sediment ??? Missing citation ???

Line 286 - For example, the content of available cadmium in soil decreased significantly by approximately 40% following the application of organic fertilizer [43] - What exactly mean ”available” in this case ?

Line 293 – should be: rare earth elements

Line 295 - Our findings indicated that residual was the mainly fraction of both Ni and Cu, accounting for 63.74% and 72.90% …  these values are the averages for all the treatments ?

Keywords – forms ??? - maybe… heavy metal forms

Table 2 – please add the explanations under the table; available nutrients – units ? Why such a particle size distribution ???

Table 4,5 – correct  ”Exchangeable”; units ???

Fig 1, Fig. 2– should be Figure 1 and Figure 2

Conclusions – in this part don’t replicate the abstract…please focus on the implications of the study results

References

Almost all the references require correction according to the journal’s requirements.

Author Response

Almost 70% of the literature is in Chinese language. Considerable part of the readers will be not able to verify the citations. In my opinion, this is not acceptable. The number of that kind of papers must be reduced to 8-10. Apart from that, some minor corrections should be made:

Line 46 - which mainly because …??? I think should be… mainly because

RE: done. We have changed in our revised manuscript.

Line 61 – sediment ??? – I think should be… soil

RE: done. We have changed in our revised manuscript.

Line 101, 126, 132, 133, 139, 144, 147-150, 158 – correct the upper and lower indices

RE: done. We have changed in our revised manuscript.

Line 131 – the solution was centrifuged …time and g-force ???

RE: the solutions were centrifuged for 25 minutes.

Line 133 – should be …adjusted

RE: done. We have changed in our revised manuscript.

Line 140 - at 96… ℃ ???

RE: yes. We have changed in our revised manuscript.

Line 148 – was heated at 86℃…not 85℃ ???

RE: done. We have changed in our revised manuscript.

Line 167 - SPSS 13.0 … please provide the manufacturer, city and country

RE: done. We have changed in our revised manuscript.

Line 172, 320 - with the maximum appeared in the treatment with 5% of bentonite addition …not with 7% ???

RE: done. We have changed in our revised manuscript.

Line 228 – adroption …should be adsorption

RE: done. We have changed in our revised manuscript.

Line 243 - In this study, the content of exchangeable heavy metals was negatively correlated with pH value… correlated ??? I don’t see calculated correlations in this paper.

RE: done. We have changed in our revised manuscript.

Line 253 - Fourthly, with the increase of pH value, the concentration of the ions of iron, aluminum and magnesium in the soil solution decreased, which facilitated the adsorption of heavy metal ions by soil. - How do you know that ??? I don’t see any results regarding the ions of iron, aluminum and magnesium in the soil solution. Missing citation ???

RE: done. We have changed in our revised manuscript.

Line 256 - Finally, the product of multivalent cations and oxyhydrogen ions in soil solution increased with increasing pH value, which improved the chance of forming such heavy metal sediments.  - How do you know that ??? … sediment ??? Missing citation ???

RE: done. We have changed in our revised manuscript.

Line 286 - For example, the content of available cadmium in soil decreased significantly by approximately 40% following the application of organic fertilizer [43] - What exactly mean ”available” in this case ?

RE: here is the soilavailableCd. We have changed in our revised manuscript.

Line 293 – should be: rare earth elements

RE: done.We have changed in our revised manuscript.

Line 295 - Our findings indicated that residual was the mainly fraction of both Ni and Cu, accounting for 63.74% and 72.90% …  these values are the averages for all the treatments ?

 RE: yes.

Keywords – forms ??? - maybe… heavy metal forms

 RE: done. We have changed in our revised manuscript.

Table 2 – please add the explanations under the table; available nutrients – units ? Why such a particle size distribution ???

RE: done. We have changed in our revised manuscript.

Table 4,5 – correct  ”Exchangeable”; units ???

RE: done. We have changed in our revised manuscript.

Fig 1, Fig. 2– should be Figure 1 and Figure 2

 RE: done. We have changed in our revised manuscript.

Conclusions – in this part don’t replicate the abstract…please focus on the implications of the study results

 RE: done. We have changed in our revised manuscript.

References

Almost all the references require correction according to the journal’s requirements.

RE: done. We have changed in our revised manuscript.

Round 2

Reviewer 1 Report

This manuscript on the soil remediation with bentonite came to my desk for the second time. Here and there, the authors had made revisions of details that were earlier commented but they have not taken a comprehensive look at their manuscript. There were some fundamental issues that I raised in my earlier report but they have not been properly addressed by the authors, and therefore my recommendation remains quite negative, detailed reasons explained below.

The main problem appears in Table 4 and Table 5 and in the associated text. The authors present that the total concentration of Cu and Ni had decreased upon addition of bentonite. I can well believe that addition of bentonite into a polluted soil can immobilize the most easily soluble fractions by specific adsorption. But the problem is in particular the decrease of the residual fraction, and the decrease of the sum of fractions (i.e., the total). The authors have made an incubation experiment where no loss of material can take place through leaching, and there are no gaseous losses of Cu and Ni. The residual fraction was dissolved with very strong agents (HF, HClO4, HNO3). No part of Cu and Ni can escape dissolution with this digestion, no matter whether it is in organic or inorganic form. I wonder if the decreases of concentrations are attributable simply to the dilution effect of the polluted soil by the bentonite. The authors don’t address this issue at all even though I pointed it out last time. If they add 50% of bentonite into a 1-kg of soil, the final mass increases to 1.5 kg. Has this been taken into account by the authors at all? How much of the decrease of concentrations can be axplained by this dilution effect? Moreover, it would be highly desirable to calculate the recovery of Cu and Ni in each treatment, i.e., how much Cu and Ni were extracted as milligrams (i.e., the mass of the metals, not the concentrations). Only if the actual amounts (in milligrams) are close to each other in all treatments, can I believe that the analytical work was carried out properly. Before these issues are addressed adequately, there is no chance of getting a positive statement of this manuscript. The authors have to really concentrate on this problem because heavy metals cannot disappear from their system.

Other remarks:

The authors are still expressing their results with unrealistic accuracy throughout the paper. Please go through every single number and consider what the realistic accuracy is. Some examples follow:

L43-44: 6102.1 ->6102, 10320.9 -> 10321

Table 2: 3121.7 ->3122, 1886.4 -> 1836

Table 4 and Table 5: No decimals should be used in these Tables.

Throughout the paper, the proportions of the different fractions have been presented with two decimals. All of them should be revised to full percentages.

L117. ”a 200mm mesh sieve”: What is this actually? Do you mean micrometers, i,e, 200 µm, or 0.2 mm?

L129 and L144: explain what these mesh sizes are in millimeters.

The English language is understandable but it is not good English. Please ask a professional linguist to polish the language.   

Author Response

This manuscript on the soil remediation with bentonite came to my desk for the second time. Here and there, the authors had made revisions of details that were earlier commented but they have not taken a comprehensive look at their manuscript. There were some fundamental issues that I raised in my earlier report but they have not been properly addressed by the authors, and therefore my recommendation remains quite negative, detailed reasons explained below.
The main problem appears in Table 4 and Table 5 and in the associated text. The authors present that the total concentration of Cu and Ni had decreased upon addition of bentonite. I can well believe that addition of bentonite into a polluted soil can immobilize the most easily soluble fractions by specific adsorption. But the problem is in particular the decrease of the residual fraction, and the decrease of the sum of fractions (i.e., the total). The authors have made an incubation experiment where no loss of material can take place through leaching, and there are no gaseous losses of Cu and Ni. The residual fraction was dissolved with very strong agents (HF, HClO4, HNO3). No part of Cu and Ni can escape dissolution with this digestion, no matter whether it is in organic or inorganic form. I wonder if the decreases of concentrations are attributable simply to the dilution effect of the polluted soil by the bentonite. The authors don’t address this issue at all even though I pointed it out last time. If they add 50% of bentonite into a 1-kg of soil, the final mass increases to 1.5 kg. Has this been taken into account by the authors at all? How much of the decrease of concentrations can be axplained by this dilution effect? Moreover, it would be highly desirable to calculate the recovery of Cu and Ni in each treatment, i.e., how much Cu and Ni were extracted as milligrams (i.e., the mass of the metals, not the concentrations). Only if the actual amounts (in milligrams) are close to each other in all treatments, can I believe that the analytical work was carried out properly. Before these issues are addressed adequately, there is no chance of getting a positive statement of this manuscript. The authors have to really concentrate on this problem because heavy metals cannot disappear from their system.

RE:We have rechecked our data, and indeed the analytical work was not carried out properly. We have changed in our revised manuscript.

Other remarks:

The authors are still expressing their results with unrealistic accuracy throughout the paper. Please go through every single number and consider what the realistic accuracy is. Some examples follow:

L43-44: 6102.1 ->6102, 10320.9 -> 10321

RE: we have changed in our revised manuscript.

Table 2: 3121.7 ->3122, 1886.4 -> 1836

RE: we have changed in our revised manuscript.

Table 4 and Table 5: No decimals should be used in these Tables.

RE: we have changed in our revised manuscript.

Throughout the paper, the proportions of the different fractions have been presented with two decimals. All of them should be revised to full percentages.

RE: we have changed in our revised manuscript.

L117. ”a 200mm mesh sieve”: What is this actually? Do you mean micrometers, i,e, 200 µm, or 0.2 mm?

RE: sorry, it is 200-mesh sieve.

L129 and L144: explain what these mesh sizes are in millimeters.

RE: sorry, we have made a mistake; it is just the mesh sieve number.

The English language is understandable but it is not good English. Please ask a professional linguist to polish the language.   

RE:we have used the English language editing service in MDPI and they have improved the English.

Reviewer 2 Report

 Accept in present form

Author Response

We have rechecked our data and changed in our revised manuscript. we have used the English language editing service in MDPI and they have improved the English.

Reviewer 4 Report

References still require some correction according to the journal’s requirements (journal name must be abbreviated; don’t forget to add the space after the name). Where are the explanations under the table 2 ? I didn’t receive an answer why authors decided on such a particle size distribution. Is it related to the domestic regulations ? I don’t see any crucial changes in the lines 243, 253, 256, 286 and 293.

Author Response

References still require some correction according to the journal’s requirements (journal name must be abbreviated; don’t forget to add the space after the name).

RE: we have changed in our revised manuscript.

Where are the explanations under the table 2? I didn’t receive an answer why authors decided on such a particle size distribution. Is it related to the domestic regulations ?

RE: Table 2 was the basic properties of the polluted soil samples. It is soil particle size distribution, we divide them into three categories according to their particle size distribution, namely Coarse(>0.25mm), Fine(0.25-0.05mm), Clay+silt(<0.05mm), which are generally used in soil science, of course not related to the domestic regulations.

I don’t see any crucial changes in the lines 243, 253, 256, 286 and 293.

RE: Line 243, here we have deleted the sentence “the content of exchangeable heavy metals was negatively correlated with pH value”. Line 253, from table 3-5, we can see that with the addition of bentonite, the pH value increased while the exchangeable form decreased, therefore, the content of exchangeable heavy metals was negatively correlated with pH value.Line 256, we have added the citation here. Line 286, we have deleted this sentence here. Line 293, we have changed in our revised manuscript.

Round 3

Reviewer 1 Report

The authors have quickly responded to the previous comments and addressed many problems pointed out in the previous review. Particularly, the English language is not fine, and the results have been presented with realistic accuracy.

But one major problem remains to be solved, i.e., the results tmemselves. Now the results of Cu and Ni fractions (Tables 4 and 5) have been completely changed. Earlier, all the fractions decreased upon addition of bentonite. In this latest manuscript, the exchangeable and carbonate-bound fractions seem to dramatically decrease (as expected) while the other fractions seem to dramatically increase to the extent that the sum of fractions ("total Cu" or "total Ni") is 3.7 times and 2.8 times higher for Ni and Cu upon 70% increase of bentonite, compared to the control. The total amounts do not match with the sum of fractions. Was the total determined separately? I don't find this information from the M&M section.

What is the explanation for this large increase of certain fractions and the total concentrations? The result is completely opposite to the earlier outcome. There are serious mistakes somewhere in the calculation (or in the analytical work itself). In a serious piece of science, the results cannot be changed so dramatically just like that. The explanation of the increase of Cu and NI in the current results remains missing, as it was missing for the previous results that were quite the opposite. - There is one very unlike explanation for the increase of Cu and Ni upon addition of bentonite. The bentonite itself may contain plenty of these metals in immobile forms, but it is highly unlikely that so hign concentrations would occur. If the authors use this explanation, they have to analyze their bentonite for Cu and Ni and the results have to match with the ones presented in Tables 4 and 5.

The authors have not checked their paper to match with the new results in Tables 4 and 5. In the Conclusions (L316-318), they write as follows: " The total and respective contents of Ni and Cu in the soil showed a downward trend due to bentonite addition, and with the increasing level of bentonite addition, the decreasing trend became increasingly steep." This statement is opposite to the results presented in Tables 4 and 5.

I am not confident with the core results of this manuscript. The authors have already been offered two opportunities to correct their paper and come out with credible and convincing results but they have declined to do so. Therefore I strongly recommend rejecting this manuscript without offering an opportunity for resubmission to this journal.

Author Response

The authors have quickly responded to the previous comments and addressed many problems pointed out in the previous review. Particularly, the English language is not fine, and the results have been presented with realistic accuracy.

RE: We have carefully checked the manuscript and some minor errors have been corrected. All the changes can be seen in our revised manuscript.

But one major problem remains to be solved, i.e., the results tmemselves. Now the results of Cu and Ni fractions (Tables 4 and 5) have been completely changed. Earlier, all the fractions decreased upon addition of bentonite. In this latest manuscript, the exchangeable and carbonate-bound fractions seem to dramatically decrease (as expected) while the other fractions seem to dramatically increase to the extent that the sum of fractions ("total Cu" or "total Ni") is 3.7 times and 2.8 times higher for Ni and Cu upon 70% increase of bentonite, compared to the control. The total amounts do not match with the sum of fractions. Was the total determined separately? I don't find this information from the M&M section.

RE: yes, the total was determined separately. We have added in our revised manuscript.

What is the explanation for this large increase of certain fractions and the total concentrations? The result is completely opposite to the earlier outcome. There are serious mistakes somewhere in the calculation (or in the analytical work itself). In a serious piece of science, the results cannot be changed so dramatically just like that. The explanation of the increase of Cu and NI in the current results remains missing, as it was missing for the previous results that were quite the opposite. - There is one very unlike explanation for the increase of Cu and Ni upon addition of bentonite. The bentonite itself may contain plenty of these metals in immobile forms, but it is highly unlikely that so hign concentrations would occur. If the authors use this explanation, they have to analyze their bentonite for Cu and Ni and the results have to match with the ones presented in Tables 4 and 5.

RE: here the soil concentration was the soil-bentonite mixture. Clearly, the total mass of the heavy metals cannot change by the BA, however, the speciation and adsorption characteristics can change.

The authors have not checked their paper to match with the new results in Tables 4 and 5. In the Conclusions (L316-318), they write as follows: " The total and respective contents of Ni and Cu in the soil showed a downward trend due to bentonite addition, and with the increasing level of bentonite addition, the decreasing trend became increasingly steep." This statement is opposite to the results presented in Tables 4 and 5.

RE: We have changed in our revised manuscript.

I am not confident with the core results of this manuscript. The authors have already been offered two opportunities to correct their paper and come out with credible and convincing results but they have declined to do so. Therefore I strongly recommend rejecting this manuscript without offering an opportunity for resubmission to this journal.

RE: Thanks for the reviewers suggestions. In this manuscript, we mainly use the laboratory incubation experimen to verify that bentonite addition can have a certain impact on the speciation and mobility of heavy metals in soil. Of course, it is impossible to completely remove the heavy metals in the soil, we must plant some trees or other method to absorb the Cu and Ni. This is also our future work to select the most suitable plants for absorbing the heavy metals. But I hope our work could be published in this journal at this stage, thank you.